# Pixel-Dependent Noise Variance Stabilization for Learning-Based Denoising in Imaging Systems with Radially Symmetric Beams

**Pablo Corral-Arroyo**[1] (iD)                 PABLO.CORRALARROYO@SIEMENS-HEALTHINEERS.COM

**Fasil Gadjimuradov**[2] (iD)                 FASIL.GADJIMURADOV@SIEMENS-HEALTHINEERS.COM

**Sai Gokul Hariharan**[2] (iD)                 SAI.HARIHARAN@SIEMENS-HEALTHINEERS.COM

[1] *Siemens Healthineers, Tafernstrasse 7, 5405 Baden, Switzerland*

[2] *Siemens Healthineers, Siemensstrasse 1, 91301 Forchheim, Germany*

## Abstract

X-ray guided medical procedures may expose the patient to a non-negligible amount of radiation dose. To mitigate the dose and reduce the risk of potentially correlated health issues, it is important to optimize the radiation exposure for both the patients and clinical staff. This means that the applied radiation dose should be as low as reasonably achievable while ensuring that the required image quality is reached. UNETs have become the state-of-the-art denoising algorithms. In this article we show a preprocessing algorithm, which improves the robustness and generalization of denosing models of systems whose images have position-dependent x-ray intensity.

**Keywords:** Image denoising, UNETs, Convolutional Neural Networks

## 1. Introduction

Denoising methods based on Convolutional Neural Networks (CNN) are well-established state-of-the-art denoising image algorithms, which deliver optimal denoising levels while preserving fine physical structures and edges. This algorithm is based on a convolutional neural network, specifically a U-NET (Ronneberger et al., 2015), a kind of CNN that consists of a compression part (encoder) and expansion part (decoder). Originally designed for image segmentation, the UNET architecture has also been applied to other image processing operations such as denoising. UNETs comprise an encoder, which is a typical convolutional network using repeated convolutions with ReLU (Rectified Linear Unit) activation and a down-sampling operation that lowers the dimension of the image. This stage compresses the spatial resolution while enriching the feature representation. The decoder then reconstructs the spatial resolution through up-convolutions and concatenating high-resolution features from the contracting path and merging this information in the following channels. Recently, it has been shown that UNET-based denoising methods outperform even the most sophisticated denoising classic methods in the context of medical imaging (Hariharan et al., 2018, 2019, 2022).

We focus on a denoising algorithm based on a UNET (Hariharan et al., 2022). The algorithm is driven by the physics underlying X-ray image formation. The transformation of received X-ray photons, collected by an indirect-detectionflat-panel detector, into pixel gray values can be described as a succession of processing stages. Each stage may involve

either a quantum gain or spatial spreading (i.e., blurring due to photon dispersion during the detection process). It is assumed that this process follows a linear model (cite). Thus, an observed noise-corrupted gray value (number of counts) $V$ at row $x$ and column $y$ can be expressed as

$$V[x,y] = \beta\big(c * k_q\big)[x,y] + n_{\text{elec}} = kN + n_{\text{elec}}, \tag{1}$$

where $c$ represents the charge signal at the photodiodes, corrupted by quantum noise, convolved with the stochastic spreading function $k_q$ and scaled by the overall detector gain $\beta$ (Siewerdsen et al., 1997). The term $n_{\text{elec}}$ denotes additive electronic noise with zero mean and standard deviation $\sigma_{\text{elec}}$. The variable $N$ represents the noise-corrupted number of detected X-ray photons, and $k$ is a proportionality constant.

The total noise present in a standard Varian X-ray image can be decomposed into its elemental components, namely electronic noise $\sigma_{\text{elec}}^2$ and quantum noise $\sigma_{\text{quant}}^2$. Accordingly, the total noise variance can be written as

$$\sigma^2 = \sigma_{\text{elec}}^2 + \sigma_{\text{quant}}^2 = \sigma_{\text{elec}}^2 + \alpha\big(kN\big)_{\text{avr}}. \tag{2}$$

Aiming to produce images with uniform noise intensity in the input image across all exposure levels, Noise Variance Stabilization algorithms are applied to the input images, such as the Generalized Ascome Transform (GAT) used in (Hariharan et al., 2022). This improves denoiser performance and the learning consistency (avoid noise learning) and shows a faster generalization.

$$V'[x,y] = \frac{2}{\alpha}\sqrt{\alpha V[x,y] + \frac{3}{8}\alpha^2 - \alpha g + \sigma_{\text{elec}}^2} \tag{3}$$

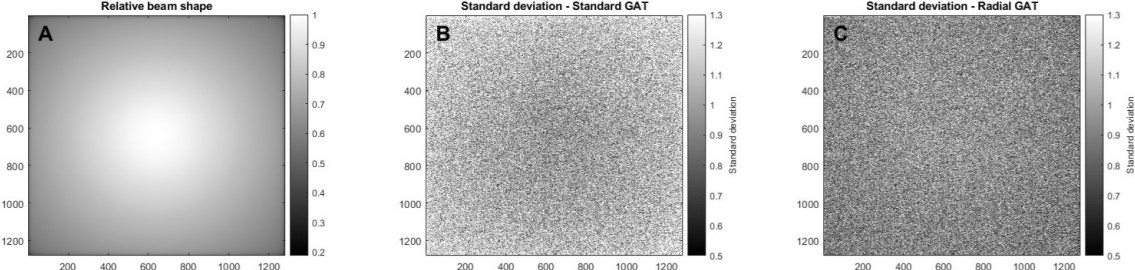

Figure 1: A) Relative beam shape map . B) Standard deviation of image after standard GAT. C) Standard deviation of image after radial GAT.

## 2. Algorithm

Typical Mega-Volts (MV) x-ray imaging system sources are based on a linear electron accelerator, after which a bending magnet directs the high-energy electrons toward the

target. When the electron beam hits the target, bremsstrahlung x-rays are produced in a forward-peaked distribution; this is a consequence of relativistic bremsstrahlung physics. The MV beam can therefore have a significant radial fall-off in the case of flattening-filter free MV beams. This relative beam shape map for a MV beam is shown in Fig. 1A, where 1 represents the maximum intensity in the central part of the beam.

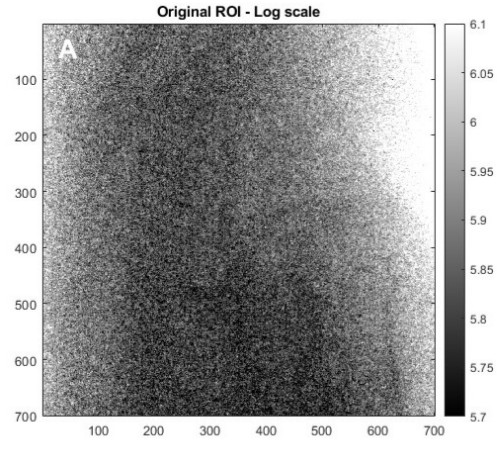
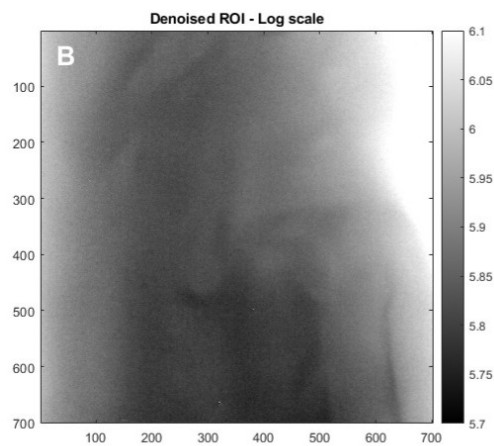

Figure 2: Noisy (A) and denoised (B) MV image of an abdomen phantom.

Due to this pronounced profile, the flat-field correction generates corrected images where the noise is amplified in the low-exposure regions and diminished in the high exposure regions. When applying a standard NVS, it is observed that the standard deviation of the noise is higher in the low-exposure regions (Fig. 1B). This poses a problem for the denoiser performance, since distinct ROIs would have different noise levels. This is solved with the radial-dependent GAT, where the noise parameters are pixel dependent accounting for the beam shape and the corresponding flat-field correction. The standard deviation is similar across the image. The application of such a pre-processing algorithm makes the training of a denoiser based on UNETs more robust for these systems across different dose levels and anatomies, while also improving signal–noise discrimination. Figure 2 shows the standard and denoised image for an abdomen phantom. The training and inference with this approach improves the generalization and robustness of the model. This same approach could be used for other similar systems.

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
