# OpenReview forum: "Pixel‑Dependent Noise Variance Stabilization for Learning‑Based Denoising in Imaging Systems with Radially Symmetric Beams"
_MIDL.io/2026/Short_Papers — MIDL 2026 - Short Papers Poster_

### Official Review · Reviewer_RDiN · 2026-05-06
**Physics-informed pixel-dependent noise stabilization for X-ray denoising; practical but limited methodological novelty**

**Rating:** 4
**Confidence:** 4

**Review:**

- This is a well-motivated and practical work grounded in imaging physics, targeting a real issue in X-ray guided imaging systems: spatially varying noise due to beam geometry and flat-panel detector response. The idea of extending noise variance stabilization to account for radial beam profiles is reasonable and naturally follows from the limitations of standard global transforms like GAT.

- The main strength of the paper is its strong physical intuition. The formulation of noise components (quantum + electronic) and the connection to beam-dependent intensity variation are clearly described. The proposed modification is also simple and computationally cheap, which is valuable for real-world deployment in medical imaging pipelines.

- However, the core idea is essentially a spatially varying extension of existing noise stabilization techniques rather than a fundamentally new learning or modeling approach. While the physics motivation is strong, the actual algorithmic contribution is incremental.

- The denoiser (U-Net) is treated as a fixed downstream model, and there is no exploration of whether the stabilization improves training dynamics, convergence, or generalization in a systematic way. The evaluation is also limited to qualitative and descriptive observations, with no quantitative comparison against standard preprocessing methods or ablations isolating the effect of radial correction.

**Summary:**

This paper proposes a pixel-dependent noise variance stabilization method for X-ray imaging systems with radially symmetric beam profiles. The method extends standard noise variance stabilization (e.g., GAT) by incorporating spatially varying exposure due to the beam shape in flat-panel detector systems. The goal is to improve robustness and generalization of U-Net-based denoisers under non-uniform noise conditions. The approach is motivated by physical modeling of X-ray photon generation and detector response, where noise variance depends on both electronic and quantum components. A radial correction is applied to stabilize noise across the image before training a U-Net denoiser. The authors demonstrate improved uniformity of noise distribution and better qualitative denoising performance on phantom abdominal images.

**Strengths:**

- Strong physics-driven motivation based on realistic X-ray imaging systems, and addresses an important practical issue on spatially varying noise in MV imaging
- Simple and computationally efficient preprocessing idea, and clear connection to established noise models (quantum + electronic noise)
- Potentially useful for improving robustness of existing U-Net denoisers and well-aligned with real-world deployment constraints in medical imaging

**Weaknesses:**

- Limited methodological novelty (extension of existing noise variance stabilization), and lack of quantitative comparison against standard preprocessing baselines
- No ablation studies isolating contribution of radial correction, also the evaluation is mostly qualitative and phantom-based, I hope the authors addressed this in final version
- Limited analysis of downstream impact on denoiser training and generalization
- Algorithmic description of radial stabilization is not fully precise
- Weak exploration of robustness across scanners, dose levels, or anatomies

**Justification Of Rating:**

The paper is strong in physical motivation and practical relevance, and the idea is useful for real X-ray denoising systems. However, from a methodological standpoint, it is an incremental extension of known noise stabilization techniques with limited experimental depth. It would be stronger with rigorous quantitative evaluation, clearer algorithmic formalization, and stronger comparison against existing preprocessing methods.

---

### Decision · Program_Chairs · 2026-05-08

Accept (Poster)